# Retrospective, multicohort analysis of the Clinical Practice Research Datalink (CPRD) to determine differences in the cost of medication wastage, dispensing fees and prescriber time of issuing either short (<60 days) or long (≥60 days) prescription lengths in primary care for common, chronic conditions in the UK

Brett Doble,[1,2] Rupert Payne,[1,3] Amelia Harshfield,[1,4] Edward C F Wilson[1]

For numbered affiliations see end of article.

**Correspondence to**
Dr Brett Doble;
brett.doble@dph.ox.ac.uk

## ABSTRACT

**Objectives** To investigate patterns of early repeat prescriptions and treatment switching over an 11-year period to estimate differences in the cost of medication wastage, dispensing fees and prescriber time for short (<60 days) and long (≥60 days) prescription lengths from the perspective of the National Health Service in the UK.

**Setting** Retrospective, multiple cohort study of primary care prescriptions from the Clinical Practice Research Datalink.

**Participants** Five random samples of 50 000 patients each prescribed oral drugs for (1) glucose control in type 2 diabetes mellitus (T2DM); (2) hypertension in T2DM; (3) statins (lipid management) in T2DM; (4) secondary prevention of myocardial infarction; and (5) depression.

**Primary and secondary outcome measures** The volume of medication wastage from early repeat prescriptions and three other types of treatment switches was quantified and costed. Dispensing fees and prescriber time were also determined. Total unnecessary costs (TUC; cost of medication wastage, dispensing fees and prescriber time) associated with <60 day and ≥60 day prescriptions, standardised to a 120-day period, were then compared.

**Results** Longer prescription lengths were associated with more medication waste per prescription. However, when including dispensing fees and prescriber time, longer prescription lengths resulted in lower TUC. This finding was consistent across all five cohorts. Savings ranged from £8.38 to £12.06 per prescription per 120 days if a single long prescription was issued instead of multiple short prescriptions. Prescriber time costs accounted for the largest component of TUC.

**Conclusions** Shorter prescription lengths could potentially reduce medication wastage, but they may also increase dispensing fees and/or the time burden of issuing prescriptions.

## Strengths and limitations of this study

► Our analysis builds on existing methodological approaches to estimate the unnecessary costs associated with different prescription lengths, providing the only evidence available from the perspective of the National Health Service in the UK.

► Limitations of our study do risk biasing the results and the reported savings (£8.38–£12.06 per prescription per 120 days) should therefore be interpreted with caution and considered upper limits.

► Clinical Practice Research Datalink (CPRD) prescription data only indicate whether a prescription has been issued and not whether it was dispensed or taken as recommended, potentially resulting in an overestimate or underestimate of the amount of wastage, depending on patient behaviour not captured in CPRD.

► The five case study conditions used in our study were purposively rather than randomly selected to represent the impact of repeat prescriptions and switching behaviour on wastage; they may not be representative of prescribing behaviour in other chronic conditions.

► Overlap of dates between prescriptions does not necessarily mean wastage has occurred and despite incorporating methods to account for this there is the possibility that our analysis approach could be overestimating the amount of medication wastage.

## INTRODUCTION

Healthcare systems worldwide are increasingly faced with the challenge of constraining rising pharmaceutical expenditures.[1] One approach to addressing this problem is to ensure prescribed medication is used as

efficiently as possible, minimising wastage. Wastage may occur when patients collect repeat prescriptions early, or when changes are made to patients' drug regimens. Intuitively, the more drugs a patient has in her/his possession at the time of a repeat prescription or regimen change, the higher the wastage. Therefore, limiting the quantity of medication through shorter prescription lengths could minimise wastage and help contain expenditure. However, the resulting higher frequency of prescriptions will have the unintended consequence of increasing transactions costs, specifically dispensing fees charged by pharmacists and healthcare professionals' time to issue them.

Several studies have examined the costs associated with issuing either long (3 months) or short (1 month) supplies of prescriptions.[2–7] In general, these concluded that shorter prescriptions were associated with lower wastage and hence reduced cost, but the increased transaction costs of shorter prescriptions more than offset these savings. These studies are all US based, which have very different healthcare systems from the UK, particularly with regard to the cost and dispensing of drugs. Therefore, the generalisability of these conclusions to the UK is questionable. Furthermore, none of the studies include healthcare professionals' time burden associated with issuing prescriptions.

In this study, we estimate differences in the costs of medication wastage and transaction costs (in terms of dispensing fees and prescriber time) in patients receiving medications within the National Health Service (NHS) in the UK as either short or long prescription lengths for five drugs/classes of drugs prescribed in primary care for common, chronic conditions.

## METHODS
### Overview
We undertook an analysis of Clinical Practice Research Datalink (CPRD)[8] prescription data to estimate the cost of medication wastage associated with shorter and longer prescription lengths for drugs used to treat five case study conditions. In order to estimate the net cost impact of shorter and longer prescription lengths, the costs of dispensing fees and prescriber time to issue a prescription were also assessed.

### Study design and inclusion criteria
This retrospective multicohort study evaluated medication wastage and its associated cost plus dispensing fees and the cost associated with issuing a prescription (ie, a general practitioner (GP) completing the process of producing a prescription; note this does not include clinical decision-making time or administrative staff time) in five condition-specific, random samples of 50 000 patients each, obtained from CPRD.

We derived the five samples from all adult patients (≥18 years old) receiving one or more prescriptions for at least one medication relevant to a case study of interest (table 1) between 1 January 2004 and 31 December 2014. In line with other studies of CPRD, data[9 10] inclusion was restricted to patients with complete data for two variables (numeric daily dose (ndd) and quantity (qty)) required

**Table 1** Case study conditions and associated prescriptions

| Case study | Relevant prescriptions/patient inclusion criteria |
|---|---|
| Glucose control with oral drug therapy in type 2 diabetes mellitus | Patients receiving one or more prescriptions for an oral antidiabetic drug listed under the BNF Section 6.1.2 Antidiabetic Drugs in any year from 2004 to 2014 |
| Treatment of hypertension in type 2 diabetes mellitus | In addition to receiving an oral antidiabetic drug as defined in (1), patients receiving one or more prescriptions for any ACE inhibitors, angiotensin II receptor antagonists, calcium-channel blockers, beta-adrenoceptor blockers, alpha-adrenoceptor blockers, potassium-sparing diuretics and/or thiazide-like diuretics in any year from 2004 to 2014 |
| Treatment with statins (lipid management) in type 2 diabetes mellitus | In addition to receiving an oral antidiabetic drug as defined in (1), patients receiving one or more prescriptions for a statin in any year from 2004 to 2014 |
| Treatment for the secondary prevention of myocardial infraction | In addition to receiving concurrent* prescriptions for an ACE inhibitor, antiplatelet and statin for at least 1 year in duration, patients receiving one or more prescriptions for beta-adrenoceptor blockers and/or angiotensin II receptor antagonists in any year from 2004 to 2014 |
| Treatment of depression | Patients receiving one or more prescriptions for any antidepressant drug listed under BNF Section 4.3 Antidepressant Drugs in any year from 2004 to 2014 |

(1) refers to the first row in the table, that is the relevant prescriptions/patient inclusion criteria for the case study "Glucose control with oral drug therapy in type 2 diabetes mellitus".
*All patients receiving at least one prescription for an ACE inhibitor, antiplatelet drug and statin were first identified in Clinical Practice Research Datalink (CPRD). Patients from this sample that did not have at least four prescriptions (chosen to represent 1 year of therapy) for each of these drugs in at least one of the 11 years of data available (ie, 2004–2014) were excluded. From the remaining patients, the additional constraint of receiving one or more prescriptions for any beta-adrenoceptor blockers and/or angiotensin II receptor antagonists was applied to define the full sample.
BNF, British National Formulary.

to calculate the prescription duration. The five case studies were defined using unique lists of product codes (CPRD unique code for treatment selected by the GP). They were: (1) glucose control with oral drug therapy in type 2 diabetes mellitus (T2DM); (2) treatment of hypertension in T2DM; (3) treatment with statins (lipid management) in T2DM; (4) treatment for the secondary prevention of myocardial infarction (MI); and (5) treatment of depression.

These were selected for study based on the chronic nature and prevalence of the associated condition within the population, the potential for a variety of prescription changes over the course of treatment and the fact that medications used in their treatment have stable dosing once therapeutic effect has been achieved, making either short or long prescription clinically appropriate.[7] Definitions of the relevant prescriptions and product code lists of the potentially prescribed medications for each of the five case studies are provided in table 1 and online supplementary appendix I, respectively. Sample data counts are provided in online supplementary appendix II.

## Treatment patterns evaluated

For each cohort, data for each patient were first ordered in sequence from earliest to latest prescription date. To identify treatment patterns, three main variables were used: (1) product code (used to identify a unique dosage, formulation and brand (or generic version) of one particular drug); (2) drug substance (used to identify different dosages and/or formulations of the same drug chemical substance); and (3) drug class (used to identify drugs with different, but related chemical composition, with similar mechanisms of action based on their categorisation in the British National Formulary). Four different prescription patterns in an individual's sequence of prescriptions were identified: (1) repeat prescriptions of the same product code; (2) substitutions between different dosages or formulations of the same drug substance; (3) substitutions between drugs that are in the same class; and (4) substitutions between drugs that have similar clinical indications from different classes. Prescriptions issued on the same day for drugs in the same class with different product codes were considered prescriber error and the duplicates were dropped from the analysis. The exception to this was for antiplatelet drugs in secondary prevention of MI, as it was assumed that two different antiplatelet drugs could be prescribed at the same time. In addition, prescriptions for medications with similar clinical indications from different classes issued on the same day were not counted as a switch, but rather as an add-on to existing therapy or concomitant therapy (online supplementary appendix II).

## Analysis of wastage

Wastage from early repeat prescriptions (pattern 1) was based on a cumulative excess supply built up over a period of 1 year. This avoided overestimation of wastage where a patient filled a prescription a few days early, but

then finished their previous supply before starting the new one. In estimating wastage from switches (patterns 2–4), we adapted a previous approach[6] to differentiate between add-ons/concomitant therapy and actual switches. If the difference between the number of changes between medications with similar clinical indications from different classes and the number of unique drug classes within a rolling annual period was ≥1, then any overlap in prescription dates was considered to be an add-on rather than a switch. This is illustrated in table 2. Similar constraints were also applied in three of the case studies (ie, the glucose control in T2DM, treatment of hypertension in T2DM and secondary prevention of myocardial infraction cohorts) due to the potential for a number of the included therapies to be given concomitantly (online supplementary appendix III).

## Costs

To estimate the costs of wastage, defined daily doses (DDD) associated with each drug substance code in the five cohorts were first obtained from the WHO's ATC/DDD Index 2016.[11] The Prescription Cost Analysis 2015, which provides details of the quantity of individual doses and net ingredient costs (NIC) of all the prescriptions in England,[12] was used to determine an NIC/quantity value of a specific strength of the medication associated with each drug substance code. This value was standardised using the associated DDD to obtain a cost per day for each drug substance code in all five of the cohorts. Details of these calculations are provided in online supplementary appendix IV.

Dispensing fees from the Drug Tariff (£0.90 per standard prescription and 2% of the cost per prescription (cost per day multiplied by prescription length) for prescriptions over £100)[13] and the estimated cost of physician or nurse time to issue a prescription were determined for each prescription. Time to issue a prescription was extracted from a targeted literature review (online supplementary appendix V). It should be noted that none of the identified studies reported times from a UK-specific primary care context. It was therefore necessary to prioritise the use of available evidence based on studies with the largest sample sizes and those studies reporting prescriber time for different types of prescriptions. Repeat prescriptions were assigned a shorter time compared with changes in dose/formulation, within drug classes and between drug classes (48.7 vs 61.2 s).[14] Per minute costs related to GPs' time (£3.80/min) or a general practice nurse's time (£0.93/min) were then applied.[15] All costs are reported in 2015 GBP.

## Statistical analysis

Descriptive analyses of trends in treatment switching and early repeat prescriptions were used to assess medication wastage. The proportion of days' supply wasted, mean number of days' supply wasted and the mean costs of wastage per prescription were determined for two prescription lengths (<60 and ≥60 days, hereafter

**Table 2** Example of differentiating between treatment switches and add-ons for a patient receiving medications for hypertension

| Year | Sequence of prescriptions in year | Drug | Class | Total number of treatment switches between classes in year (A) | Total number of unique classes in year (B) | Difference for year (A)–(B) | Count as treatment switch between classes* | Count as add-on |
|---|---|---|---|---|---|---|---|---|
| 2011 | 1 | Ramipril | ACE | 1 | 2 | −1 | No | No |
| 2011 | 2 | Losartan potassium | ARA | | | | Yes | No |
| 2011 | 3 | Losartan potassium | ARA | | | | No | No |
| 2011 | 4 | Losartan potassium | ARA | | | | No | No |
| 2012 | 1 | Losartan potassium | ARA | 2 | 2 | 0 | No | No |
| 2012 | 2 | Diltiazem hydrochloride | CCB | | | | No | Yes |
| 2012 | 3 | Diltiazem hydrochloride | CCB | | | | No | No |
| 2012 | 4 | Losartan potassium | ARA | | | | No | Yes |
| 2013 | 1 | Losartan potassium | ARA | 4 | 2 | 2 | No | No |
| 2013 | 2 | Doxazosin | AAB | | | | No | Yes |
| 2013 | 3 | Losartan potassium | ARA | | | | No | Yes |
| 2013 | 4 | Doxazosin | AAB | | | | No | Yes |
| 2013 | 5 | Losartan potassium | ARA | | | | No | Yes |

*For the treatment of hypertension in type 2 diabetes mellitus (T2DM) cohort, overlaps in prescription dates involving ACE inhibitors and angiotensin II receptor antagonists with either calcium-channel blockers or thiazide-like diuretics were not counted as switches as these therapies are commonly administered together as second-line therapy.[20]

AAB, alpha-adrenoceptor blocker; ARA, angiotensin II receptor antagonist; CCB, calcium-channel blocker.

In 2011, the patient has one change between clinically related drugs from different classes (ramipril to losartan) and receives medication belonging to two unique drug classes (ACE and ARA). One minus two is <1, so this change is considered a switch. The rationale being that if the number of changes was small or large and the number of unique drugs involved in the changes was also small or large, respectively, switches in therapies were occurring and therefore there was potential for wastage to occur. In 2012, the patient has two changes between clinically related drugs from different classes (losartan to diltiazem and diltiazem to losartan) and receives medication belonging to two unique drug classes (ARA and CCB). Two minus two is <1, which indicates a switch, but in the treatment of hypertension, ARAs and CCBs are commonly administered together as second-line therapy[20] and therefore these two changes were considered add-ons/concomitant therapy. In 2013, the patient has four changes between clinically related drugs from different classes (losartan to doxazosin, doxazosin to losartan, losartan to doxazosin, and doxazosin to losartan) and receives medication belonging to two unique drug classes (ARA and AAB). Four minus two is ≥1, which indicates the four changes are add-ons, not switches. The rationale being that if the number of changes was large, but the number of unique drugs involved in the changes was low, an add-on or concomitant therapy was being prescribed and no wastage was occurring.

'short' and 'long' prescriptions) over the 11-year period. Mean cost of wastage per prescription was reported for each of the four treatment patterns individually and for all treatment patterns combined for each annual period. Two-sample t-tests using groups (<60 and ≥60 day prescription lengths) assuming unequal variance were used to compare the differences between the <60 day and ≥60 day groups.

To determine and compare the total unnecessary costs (TUC; cost of medication wastage, dispensing fees and prescriber time) associated with short and long prescription lengths, a model originally used by Walton *et al*[b] was adapted and applied to the prescription data from the five cohorts (online supplementary appendix VI), and the two equations below were used, where 'C' represents cost and 'Q' represents quantity. An example of calculation is provided in box.

$$TUC_{<60} = (C_{wastage<60} + C_{dispensing<60} + C_{prescribertime<60})$$
$$\times (120/Q_{daysused<60}) - (C_{dispensing<60} + C_{prescribertime<60}) \quad (1)$$

$$TUC_{60} = (C_{wastage60} + C_{dispensing60} + C_{prescribertime60})$$
$$\times (120/Q_{daysused60}) - (C_{dispensing60} + C_{prescribertime60}) \quad (2)$$

One-way sensitivity analyses were conducted to examine differences in TUC under a variety of different scenarios, including scenarios assuming nurses issued

**Box  Example comparing total unnecessary costs for <60 day and ≥60 day prescription lengths for a standardised time period of 120 days**

Assume on average that the <60 day prescription length is 35 days and the average ≥60 day prescription length is 120 days. Also assume that regardless of prescription length, patients on average switch their prescription 30 days after a prescription is issued. The quantity used is therefore 30 days for both prescription lengths ($Q_{days\,used<60}$ = $Q_{days\,used≥60}$ = 30), but the quantity wasted is much larger for the ≥60 day prescription (90 days compared with only 5 days wasted). Since over a 120-day period both prescription lengths will incur the same dispensing fees and prescriber time costs (four prescriptions will be issued regardless of prescription length as a switch occurs every 30 days), the ≥60 day prescription will be associated with higher total unnecessary costs. Note that this example has been developed by adapting an example provided by Walton *et al*[b] to the prescription lengths considered in our study.

the prescription instead of a GP, excluding prescriber time costs, accounting for changes in NHS revenue from patient charges per prescription, ±50% mean days wasted, ±50% the mean cost of drugs per day, dispensing fees and prescriber time.

All statistical analyses were performed using Stata/MP V.13.1 (StataCorp, College Station, Texas, USA).

## RESULTS

### Overall cohort selection

The proportion of observations dropped from the full sample due to missing or observations equal to zero in either the ndd or qty variables ranged from 6% in both the lipid management and hypertension cohorts to 21% in the glucose control in T2DM cohort. The numbers of observations were further reduced after accounting for prescription error (online supplementary appendix II).

### Medication wastage

Over the 11-year study period, there was a statistically significant difference in the proportion of days' supply wasted, mean number of days' supplied wasted and the mean cost of wastage per prescription between the short and long prescription groups for all five of the case studies (online supplementary appendix VIII). The proportion of days' supply wasted was consistently larger for the long prescription group across all cohorts except depression where the short group had 6.3% of days' supply wasted compared with 3.7% in the longer group. The mean number of days' supply wasted was also consistently larger for the longer group, but the difference between the two prescription length groups was much smaller for the depression cohort in comparison to the other four cohorts.

### Medication wastage by treatment pattern

In four of the five cohorts, mean cost of wastage per prescription was significantly higher with longer prescription lengths for all four treatment patterns (table 3). The one exception was for the depression cohort where the mean cost of wastage per prescription for both dosage/formulation and within-class treatment switches did not show statistically significant differences between the two prescription length groups. The repeat prescription treatment pattern consistently had the largest mean cost of wastage per prescription across the cohorts, particularly for the longer groups, except for the depression cohort. The lipid management cohort did not report any between-class treatment switches as all medications included in the analysis were from the same class of statins.

### Medication wastage over time

On an annual basis, mean cost of wastage per prescription was significantly higher in the longer prescription lengths for each study year, except 2012 and 2013 for depression (online supplementary appendix IX). In general, the magnitude of the mean costs remained relatively consistent over the study period, except for a few notable

**Table 3** Comparison of the mean cost of medication wastage per prescription over a 11-year period (2004–2014) by treatment pattern (2015 £)

| | Mean cost of repeat prescription wastage per prescription 2015 £ (95% CI) | | Mean cost of dosage/formulation switch wastage per prescription | | Mean cost of within-class treatment switch wastage per prescription 2015 £ (95% CI) | | Mean cost of between-class treatment switch wastage per prescription 2015 £ (95% CI) | |
|---|---|---|---|---|---|---|---|---|
| | <60 days | ≥60 days | <60 days | ≥60 days | <60 days | ≥60 days | <60 days | ≥60 days |
| Glucose control with oral drug therapy in T2DM | 0.230 (0.226 to 0.233) | 1.035 (0.772 to 1.298) | 0.059 (0.058 to 0.061) | 0.173 (0.143 to 0.204) | 0.031 (0.029 to 0.032) | 0.097 (0.080 to 0.114) | 0.009 (0.008 to 0.010) | 0.064 (0.051 to 0.078) |
| Hypertension in T2DM | 0.050 (0.049 to 0.051) | 0.271 (0.228 to 0.314) | 0.038 (0.038 to 0.039) | 0.128 (0.107 to 0.149) | 0.004 (0.003 to 0.004) | 0.013 (0.009 to 0.016) | 0.003 (0.003 to 0.003) | 0.026 (0.022 to 0.030) |
| Lipid management in T2DM | 0.017 (0.017 to 0.017) | 1.099 (0.832 to 1.367) | 0.024 (0.023 to 0.024) | 0.153 (0.081 to 0.225) | 0.008 (0.007 to 0.008) | 0.173 (0.075 to 0.271) | NA | NA |
| Secondary prevention of myocardial infraction | 0.043 (0.042 to 0.043) | 0.439 (0.300 to 0.578) | 0.014 (0.014 to 0.014) | 0.040 (0.036 to 0.045) | 0.009 (0.009 to 0.009) | 0.029 (0.027 to 0.031) | 0.00005 (0.00004 to 0.00006) | 0.0006 (0.0003 to 0.0008) |
| Depression | 0.044 (0.042 to 0.046) | 0.214 (0.180 to 0.249) | 0.146 (0.143 to 0.150) | 0.141 (0.113 to 0.169) | 0.006 (0.005 to 0.007) | 0.013 (0.004 to 0.021) | 0.012 (0.010 to 0.013) | 0.061 (0.036 to 0.086) |

NA, not applicable; T2DM, type 2 diabetes mellitus.

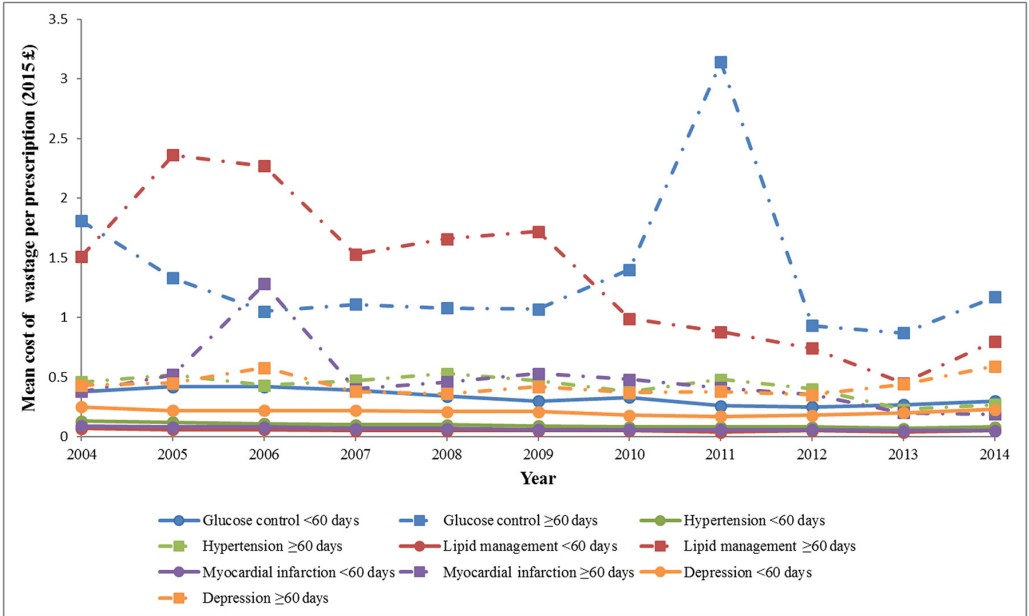

**Figure 1** Trends in the mean cost of medication wastage per prescription over 11-year study period.

trends (figure 1). In the glucose control in T2DM cohort, the mean costs for the longer group in years 2004 and 2011 were slightly larger (range £1.81–£3.14) compared with the other nine annual means (range £0.87–£1.40). In the hypertension cohort, there was a slight trend of decreasing magnitude of the mean cost over the 11 years for the shorter group; a decrease in mean cost was limited to years 2013 and 2014 in the longer group. For both the lipid management and secondary prevention of MI cohorts, the magnitude of the mean costs remained relatively consistent over the 11 years for the shorter prescription length groups, whereas there was a slightly decreasing trend in the magnitude of the mean costs for the longer prescription length groups.

### Differences in TUCs for short and long prescription lengths

TUC (wastage, dispensing fees and prescriber time) per 120 days was lower in the longer prescription group for all five cohorts (savings of £8.38 (glucose control in T2DM) to £12.06 (secondary prevention of MI) per prescription per 120 days if a single long prescription was issued instead of multiple short prescriptions, online supplementary appendix X). This roughly translates into savings of £25.14–£36.18 per patient per year assuming patients would receive three prescriptions each with a 120-day supply instead of 12 prescriptions each with a 30-day supply.

Sensitivity analysis shows longer prescriptions remained cost saving compared with shorter prescriptions across all scenarios and ranges tested. The magnitude of the savings was lowest when prescriber time costs were excluded from the models (range £0.91–£2.81 per prescription per 120 days) and reduced to a lesser extent when nurse prescriber time costs were used instead of physician's (range £5.94–£8.48 per prescription per 120 days) and when loss of revenue to the NHS through a reduced number of

prescription charges paid by patients was incorporated into the models (range £6.52–£9.83 per prescription per 120 days). The other scenarios tested had relatively little impact on the magnitude of the savings, with the exception of increases and decreases of 50% in the cost of prescriber time (online supplementary appendix X).

## DISCUSSION
### Summary of findings

Longer prescription lengths are associated with more medication wastage per prescription compared with shorter prescription lengths. However, after taking into account transaction costs, longer prescription lengths are associated with overall cost savings (lower TUC) compared with shorter ones. In all five cohorts, most prescriptions were for ≤30 days with relatively small proportions of patients having prescription lengths between 31 and 60 days (18%, 27%, 28%, 27% and 25% for the depression, T2DM, hypertension, lipid management and MI cohorts, respectively). Ninety-five per cent of prescriptions in the depression cohort were for <60 days. Some 39 million prescriptions are issued for antidepressants in the UK each year,[16] therefore, if the 95% issued as <60 day supplies was instead issued as longer ≥60 day prescriptions the total savings to the NHS could be as much as £408 million per year. Similarly, knowing 97.05% of statin prescriptions were issued as <60 day prescriptions from our CPRD analysis, the total savings to the NHS just in England could be as much as £563 million per year if the ~61.1 million[17] short statin prescriptions issued in 2015 for two statins (simvastatin and atorvastatin) were changed to longer prescriptions. However, it is critical to note that the majority of savings for both examples will not be cash releasing, but will be realised as savings in GP time, which

could be used to increase primary care consultations with patients. Cash-releasing savings may come from reduced dispensing fees, for which we estimate an upper limit of £104 million and £62 million for antidepressants and the statins, respectively. However, these cash savings will come at the expense of community pharmacies that may rely on dispensing fees to support their businesses, a fact that should be considered if longer prescription lengths are to be adopted in practice. The magnitude of the savings for the other case studies will be of a similar scale given the prevalence of the conditions and frequency of shorter prescriptions. These figures should be interpreted with caution as they assume it is clinically appropriate for all prescriptions to be issued for a longer duration, which will certainly not be the case.

### Comparison to previous studies

Several other studies have examined the costs associated with issuing either long (3 months) or short (1 month) supplies of prescriptions.[2–7] These studies all take the perspective of various payers in the USA (eg, different state-level Veterans Affairs and Medicaid programmes as well as a non-institutionalised civilian population) and account for different cost items. Two studies found savings associated with longer prescriptions of a similar magnitude to ours, for example, TUC of US$2.45 (£1.63 at April 2015 exchange rates)[5] and US$6.17 (£4.10).[3] The former study[5] excluded prescriber time (the equivalent figure in our study is £1.03), and the latter[3] included costs of mail-order prescriptions.

Another study calculated per patient per year savings of US$7.70 (statins) to US$26.86 (oral hypoglycaemics) associated with 90-day vs 30-day prescriptions.[6] A study of the financial impact on healthcare payers[4] detected statistically significant savings with 3-month supplies in only two of six cases as most savings accrued to patients through reductions in out-of-pocket costs. This study did not consider the cost of medication wastage making comparison with our study difficult. A simulation study found that any savings from reduced wastage from a shorter prescription length were more than offset by increases in dispensing fees as long as the dispensing fee was at least US$2.40 (base case assumption was US$5.60).[2]

In contrast to these, a comprehensive study on the impact of a policy to reduce the maximum prescription length from 100 to 34 days' supply in the North Carolina Medicaid programme[7] found that total Medicaid expenditures (comprising outpatient, inpatient, emergency as well as pharmacy costs) decreased for patients initially receiving 100-day prescriptions after the implementation of the 34-day policy (range US$245–US$440 per person per quarter across six classes of medications (antihypertensives, antidiabetic medications, lipid-lowering drugs, seizure-disorder medications, antidepressants and antipsychotics) assessed). However, the results are not broken down by expenditure category (except for reporting decreases in expenditures for the targeted prescriptions across all six medication classes) and

therefore it is unclear where the savings are accrued. This finding may be explained by small adverse health effects as a result of changes in adherence, patients absorbing any health effects through informal care or tolerating greater disease burden, or the follow-up period of the study (18 months after implementation) being too short to capture any spillover effects of decreased adherence on other Medicaid services. The equivalent impact on NHS expenditure in the UK may differ due to differences in the organisation of care, in particular the gate-keeper role of primary care. Analysis of this was outside the scope of our analysis but would be a valuable future line of enquiry.

### STUDY LIMITATIONS

To the best of the authors' knowledge, this study provides the only evidence of the unnecessary costs associated with different prescription lengths from the perspective of the NHS in the UK and builds on existing methodological approaches available in the literature. However, there are a few limitations that warrant discussion. First, CPRD prescription data only indicate whether a prescription has been issued and not whether it was dispensed or taken as recommended. Our estimates may, therefore, either overstate or understate the amount of wastage that actually occurred, depending on patient behaviour not captured in CPRD.

Second, the five case study conditions were purposively rather than randomly selected to represent the impact of repeat prescriptions and switching behaviour on wastage; they may not be representative of prescribing behaviour in other chronic conditions. However, those selected do represent some of the most common chronic conditions treated with prescribed medications. Nine of the top 20 prescribed medications within NHS England were included in at least one of the case study conditions in our analyses, and combined they accounted for around £378 million (4%) of all drug expenditures within NHS England in 2015 and are therefore highly policy relevant.[17] Our analysis also excluded patients having one or more observations with missing or zero values for either the ndd and/or qty variables. If this was non-random then the subsequent samples may not be truly representative of the general population. Appropriate methods to impute these variables are of limited value and our approach was similar to other studies using CPRD data.[9 10]

Third, the identification of patients within CPRD for the five case studies (table 1) was based solely on product codes, rather than in conjunction with medical diagnoses. It is therefore possible that some of the patients in the five cohorts may be receiving medications for other conditions not of specific interest in our study (eg, antidepressants used for anxiety or metformin used for polycystic ovary disease). However, as the main aim of our study was to estimate drug wastage, the possible inclusion of patients with conditions outside our cohort definitions still provided our analysis with relevant information

concerning drug wastage, dispensing fees and prescriber time.

Fourth, an overlap of dates between prescriptions does not necessarily mean wastage has occurred as consumption of early repeat prescriptions may be delayed until the initial supply is exhausted and treatment changes might actually be add-ons to existing prescriptions or concomitant therapy rather than switches in therapy. To ensure wastage was not overestimated, a threshold of 1 year after the initial prescription in a particular series was used to estimate wastage for early repeat prescriptions, and a threshold of <1 in the difference between the number of drug changes between medications with similar clinical indications from different classes and the number of unique drug classes within an annual period was used to identify wastage from between-class treatment switches. There is, however, the possibility that our analysis approach could be overestimating the amount of medication wastage.

Fifth, for pragmatic purposes we dichotomised prescription lengths into 'short' versus 'long', with a cut-off of 60 days. This will have classified 56-day prescriptions as 'short'. While this will have resulted in a loss of sensitivity (there may be differences in TUC between 1 and 2-month prescriptions), the overall conclusions comparing 'shorter' (<60 days) and 'longer' (≥60 days) lengths are not affected.

Finally, a number of assumptions were required to assign unit costs to the estimated proportions of wastage. Mean cost per day values derived using DDDs, NICs and quantities at the drug substance level were calculated and then applied to any prescription categorised under that particular drug substance. This approach is not ideal, but necessary given the inability to link CPRD data to individual unit costs specific for each prescription. The direction and magnitude of any resulting bias are difficult to predict.

Furthermore, NICs do not include any discounts that may be applied or include any adjustment for revenue received by the NHS where a prescription charge is paid at the time the prescription is dispensed or where the patient has purchased a prepayment certificate, and therefore may be different from the net cost incurred specifically by the NHS. Patients with T2DM are exempt from the prescription charge,[18] and overall almost 90% of prescriptions dispensed in the NHS in England are exempt.[19]

All these limitations risk biasing the results. The projected savings should therefore be interpreted with caution and in any case be considered upper limits. Our analysis focused on drugs with low unit costs prescribed to large numbers of patients. The results may not be generalisable to high-cost drugs used to treat relatively small patient groups.

## CONCLUSIONS

Overall, the findings from the study indicate that from the perspective of the NHS in the UK, longer prescription lengths are cost-saving relative to shorter prescription lengths in a number of common chronic diseases. Policymakers should recommend that GPs consider issuing longer prescriptions for common chronic conditions where clinically appropriate to minimise the costs associated with dispensing fees and prescriber time as a result of issuing multiple prescriptions of shorter length.

**Author affiliations**
[1]Cambridge Centre for Health Services Research, University of Cambridge, Cambridge, UK
[2]Health Economics Research Centre, Nuffield Department of Population Health, University of Oxford, Oxford, UK
[3]Department of Population Health Sciences, Bristol Medical School, Centre for Academic Primary Care, University of Bristol, Bristol, UK
[4]Department of Public Health and Primary Care, University of Cambridge, Cambridge, UK

**Acknowledgements** The authors acknowledge the valuable feedback received from Céline Miani, Adam Martin, Josephine Exley, Catherine Meads and Sarah King during the conduct of this study.

**Contributors** BD designed the study protocol; extracted, analysed and interpreted the data; drafted and revised the article; and gave final approval of the version to be published. AH extracted and assisted in organising the data; reviewed and edited the draft article; and gave final approval of the version to be published. RP and ECFW conceptualised the study; assisted with its design and the interpretation of data; critically reviewed and edited the draft article; and gave final approval of the version to be published.

**Funding** This research was supported by a grant from the National Institute for Health Research, Health Technology Assessment funding stream (Grant reference: NIHR HTA 14/159/07). This article presents independent research funded by the National Institute for Health Research (NIHR). The views expressed are those of the authors and not necessarily those of the National Health Service, the National Institute for Health Research or the Department of Health.

**Disclaimer** This study is based on data from the Clinical Practice Research Datalink obtained under license from the UK Medicines and Healthcare Products Regulatory Agency. The interpretation and conclusions contained in this study are those of the authors alone.

**Competing interests** None declared.

**Ethics approval** The protocol (16_117R) forthis study was approved on 21 June 2016 bythe Independent Scientific Advisory Committee (ISAC), the independentbody that approves use of CPRD data (online supplementary appendix VII).

**Provenance and peer review** Not commissioned; externally peer reviewed.

**Data sharing statement** No additional data available.

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
