## [Reviewer comments · BMJ Open]

ARTICLE DETAILS

TITLE (PROVISIONAL)	A retrospective, multi-cohort analysis of the Clinical Practice Research Datalink (CPRD) to determine differences in the cost of medication wastage, dispensing fees and prescriber time of issuing either short (<60 day) or long (≥60 day) prescription lengths in primary care for common, chronic conditions in the United Kingdom
AUTHORS	Doble, Brett; Payne, Rupert; Harshfield, Amelia; Wilson, Edward

VERSION 1 – REVIEW

REVIEWER	Henrik Støvring Department of Public Health, Biostatistics Aarhus University Denmark
REVIEW RETURNED	21-Sep-2017

GENERAL COMMENTS	The paper presents results based on an analysis of data from a large and validated registry (CPRD) pertaining to an important public health questions: Should GPs from an economic perspective prefer issuing short or long prescriptions in treatment of chronic conditions? The main finding of the paper is that while medication wastage may be larger for longer prescriptions, the cost is offset by reductions in costs related to issuing and filling the prescription. The paper is based on a pre-approved protocol for the analysis, which is a major strength. The description of methods and results is transparent and clear, and the discussion is thorough with respect to the limitations. I have a couple of comments to the paper for the authors too consider: Defining the duration of a prescription is the cornerstone in this study, as it is in most pharmacoepidemiological studies. Authors define it from information from the issuing doctor regarding amount dispensed and dosage to be taken each day. This information is not complete for all prescriptions, as described in the paper. I think more information should be provided on the characteristics of patients, where the information is incomplete, as this would benefit assessment of the generalizability of results. Further, while the defined duration accurately reflects the state of mind of the doctor at the time of issuance, I wonder whether anything is known regarding doctor initiated dosage change while patients use a prescription? If this happens, authors likely do not have information on it in their database, but I wonder whether it then would make sense to make a sensitivity analysis regarding for example the effect of having
---

	defined durations as 5% too short? Or 5% too long? Authors acknowledge that choice of prescription length may depend on clinical considerations. I think this is a crucial point, as it in essence highlights the challenge of using results from an observational study to recommend interventions. In principle, 100% of issued prescriptions could have the "correct" length from a clinical perspective, in which case this analysis seems moot. Are there any studies or supporting evidence that suggests actual substitution of short prescriptions with long prescription is possible for a substantial group of patients?
--	--

REVIEWER	Renaudin Pierre Université de Montpellier Laboratoire de Pharmacie Clinique UFR Pharmacie France
REVIEW RETURNED	28-Sep-2017

GENERAL COMMENTS	Very well written article and clear. Very interesting question is that this is not extrapolable to other countries. Nothing to change
---

VERSION 1 – AUTHOR RESPONSE

Reviewer: 1

Reviewer Name: Henrik Støvring

Institution and Country: Department of Public Health, Biostatistics, Aarhus University, Denmark

Please state any competing interests: None declared.

1. The paper presents results based on an analysis of data from a large and validated registry (CPRD) pertaining to an important public health questions: Should GPs from an economic perspective prefer issuing short or long prescriptions in treatment of chronic conditions? The main finding of the paper is that while medication wastage may be larger for longer prescriptions, the cost is offset by reductions in costs related to issuing and filling the prescription. The paper is based on a pre-approved protocol for the analysis, which is a major strength. The description of methods and results is transparent and clear, and the discussion is thorough with respect to the limitations. I have a couple of comments to the paper for the authors too consider:

Authors' response: Thank you for the positive feedback, responses to your comments are provided below.

2. Defining the duration of a prescription is the cornerstone in this study, as it is in most pharmacoepidemiological studies. Authors define it from information from the issuing doctor regarding amount dispensed and dosage to be taken each day. This information is not complete for all prescriptions, as described in the paper. I think more information should be provided on the characteristics of patients, where the information is incomplete, as this would benefit assessment of the generalizability of results.

Authors' response: Unfortunately we only had access to the 'therapy' file from CPRD for our analysis and therefore we did not have any information on general demographics or other patient

characteristics for any of the patients either included or excluded from our analyses (i.e., the data we had was limited to prescription information only). We were however, able to identify what drugs were most commonly associated with with qty and ndd variables equal to zero. We have added this information, as below, to the footnotes in Appendix II that describes the data processing of the five cohorts:

“aThe product codes that most frequently had qty or ndd values equal to zero were: metformin 500 mg tablets (38%) and gliclazide 80 mg tablets (26%) for the T2DM cohort; ramipril 10 mg capsules (9%), bendroflumethiazide 2.5 mg tablets, Ramipril 5 mg capsules, Ramipril 2.5 mg capsules and amlodipine 5 mg tablets for the hypertension in T2DM cohort; simvastatin 40 mg tablets (38%), simvastatin 20 mg tablets (16%), atorvastatin 40 mg tablets (10%) and atorvastatin 20 mg tablets (10%) for the lipid management in T2DM cohort; aspirin 75 mg dispersible tablets (21%) and simvastatin 40 mg tablets (12%) for the secondary prevention of myocardial infarction cohort; and citalopram 20 mg tablets (13%) and fluoxetine 20 mg capsules (11%) for the depression cohort.”

3. Further, while the defined duration accurately reflects the state of mind of the doctor at the time of issuance, I wonder whether anything is known regarding doctor initiated dosage change while patients use a prescription? If this happens, authors likely do not have information on it in their database, but I wonder whether it then would make sense to make a sensitivity analysis regarding for example the effect of having defined durations as 5% too short? Or 5% too long?

Authors' response: It is not clear why such a sensitivity analysis would be important to facilitate the interpretation of our conclusions. All changes in dosages were captured in our analysis whether they were doctor initiated or patient initiated and it is not clear why the impact of doctor initiated changes are more important to consider. Our study is meant to be an empirical analysis that uses real world data and not a simulation study to test hypothetical scenarios as proposed by the reviewer. In addition, such sensitivity analyses were not pre-specified in the approved protocol and therefore we have decided not to conduct the suggested sensitivity analyses as proposed by the reviewer.

4. Authors acknowledge that choice of prescription length may depend on clinical considerations. I think this is a crucial point, as it in essence highlights the challenge of using results from an observational study to recommend interventions. In principle, 100% of issued prescriptions could have the "correct" length from a clinical perspective, in which case this analysis seems moot. Are there any studies or supporting evidence that suggests actual substitution of short prescriptions with long prescription is possible for a substantial group of patients?

Authors' response: Short prescription lengths are actually a relatively new policy and historically a large number of prescriptions assessed in our study were given as long prescriptions lengths. This is evidenced by the study conducted by Domino et al. (reference number 7 in our study) that assessed the impact of a policy to reduce the maximum prescription length from 100 to 34 days' supply in the North Carolina Medicaid program. Six medication categories used to treat chronic conditions: anti-hypertensives, anti-diabetic medications, lipid-lowering medications, seizure medications, antidepressants and antipsychotics were assessed by Domino et al. and were used by us to help define appropriate case study conditions for our study where prescriptions could be issued as either short or long prescription lengths. These specific conditions were selected by Domino et al. as medication for these conditions have "stable dosing once therapeutic effect has been achieved." and therefore would be considered clinically appropriate to be issued as either long or short prescription lengths.

The below text was added to the methods (page 6) to justify the selection of our five case study conditions and highlight reasons why either short or long prescriptions would be clinically appropriate for medications used to treat these conditions.

“These were selected for study based on the chronic nature and prevalence of the associated condition within the population, the potential for a variety of prescription changes over the course of treatment and the fact that medications used in their treatment have stable dosing once therapeutic effect has been achieved, making either short or long prescription clinically appropriate.

Reviewer: 2

Reviewer Name: Renaudin Pierre

Institution and Country: Université de Montpellier, Laboratoire de Pharmacie Clinique, UFR Pharmacie, France

Please state any competing interests: None declared

1. Very well written article and clear. Very interesting question is that this is not extrapolable to other countries. Nothing to change.

Authors' response: Thank you for the positive feedback.

VERSION 2 – REVIEW

REVIEWER	Henrik Støvring Department of Public Health Aarhus University
REVIEW RETURNED	12-Oct-2017
GENERAL COMMENTS	Authors have adequately responded to the points I have raised. Nothing further to add from me.